# Conductance stable and mechanically durable bi-layer EGaIn composite-coated stretchable fiber for 1D bioelectronics

Gun-Hee Lee [1,2,3,9], Do Hoon Lee[1,9], Woojin Jeon [3,9], Jihwan Yoon [4], Kwangguk Ahn[4], Kum Seok Nam [3], Min Kim[1], Jun Kyu Kim[1], Yong Hoe Koo[5], Jinmyoung Joo [5], WooChul Jung [1], Jaehong Lee [6], Jaewook Nam[4], Seongjun Park [3,7,8] ✉, Jae-Woong Jeong [2,7] ✉ & Steve Park [1,7,8] ✉

Deformable semi-solid liquid metal particles (LMP) have emerged as a promising substitute for rigid conductive fillers due to their excellent electrical properties and stable conductance under strain. However, achieving a compact and robust coating of LMP on fibers remains a persistent challenge, mainly due to the incompatibility of conventional coating techniques with LMP. Additionally, the limited durability and absence of initial electrical conductivity of LMP restrict their widespread application. In this study, we propose a solution process that robustly and compactly assembles mechanically durable and initially conductive LMP on fibers. Specifically, we present a shearing-based deposition of polymer-attached LMP followed by additional coating with CNT-attached LMP to create bi-layer LMP composite with exceptional durability, electrical conductivity, stretchability, and biocompatibility on various fibers. The versatility and reliability of this manufacturing strategy for 1D electronics are demonstrated through the development of sewn electrical circuits, smart clothes, stretchable biointerfaced fiber, and multifunctional fiber probes.

The burgeoning interest in electrically conductive fibers can be attributed to the growing demand for smart wearable clothing, smart sutures, and minimally invasive bioelectronics[1–7]. To ensure seamless integration of conductive fibers with biological interfaces, the mechanical attributes of softness and stretchability play a pivotal role[6–9]. However, traditional metal wire-based conductive fibers are not compatible with soft biointerfaces and are susceptible to fatigue

fracture[2,6]. To circumvent this challenge, a wide range of investigations have focused on the coating of soft and stretchable fibers with conductive fillers such as carbon nanotubes (CNTs), graphene, and silver nanowires/particles to incorporate electrical functionality[10–14]. Nevertheless, solid conductive filler-coated fibers display a noticeable increase in electrical resistivity under strain, with a high gauge factor[2,10,11,13,14]. This collapse of electrical conductivity is a fundamental

[1]Department of Materials Science and Engineering, Korea Advanced Institute of Science and Technology (KAIST), 291 Daehak-ro, Yuseong-gu, Daejeon 34141, Republic of Korea. [2]School of Electrical Engineering, Korea Advanced Institute of Science and Technology (KAIST), 291 Daehak-ro, Yuseong-gu, Daejeon 34141, Republic of Korea. [3]Department of Bio and Brain Engineering, Korea Advanced Institute of Science and Technology (KAIST), 291 Daehak-ro, Yuseong-gu, Daejeon 34141, Republic of Korea. [4]School of Chemical and Biological Engineering, Institute of Chemical Processes, Seoul National University, 599 Gwanak-ro, Gwanak-gu, Seoul 08826, Republic of Korea. [5]Department of Biomedical Engineering, Ulsan National Institute of Science and Technology (UNIST), 50, UNIST-gil, Ulju-gun, Ulsan 44919, Republic of Korea. [6]Department of Robotics and Mechatronics Engineering, Daegu Gyeongbuk Institute of Science and Technology (DGIST), 333 Techno Jungang-daero, Daegu 42988, Republic of Korea. [7]KAIST Institute for Health Science and Technology, 291 Daehak-ro, Yuseong-gu, Daejeon 34141, Republic of Korea. [8]KAIST Institute for NanoCentury, 291 Daehak-ro, Yuseong-gu, Daejeon 34141, Republic of Korea. [9]These authors contributed equally: Gun-Hee Lee, Do Hoon Lee, Woojin Jeon. ✉e-mail: spark19@kaist.ac.kr; jjeong1@kaist.ac.kr; stevepark@kaist.ac.kr

constraint of solid conductive fillers, predominantly controlled by percolation theory, which results in a reduction in the number of conductive pathways, and thereby, limits their application as stretchable interconnects or electrodes[15,16].

In recent times, the potential of gallium-based liquid metal particles (LMPs) as a promising conductive filler has been recognized, primarily due to their unique combination of properties such as low gauge factor, exceptional deformability with zero modulus, and high electrical conductivity (~$2.5 \times 10^6$ S/m)[17–22]. However, LMPs possess certain inherent limitations that need to be addressed for their effective utilization in fiber electronics. Firstly, LMPs coated with a thin native oxide layer exhibit electrical insulation, necessitating additional processes for electrical activation[22–25]. Secondly, the fragility of LMPs under shear force necessitates the incorporation of the additional embedding matrix to improve their durability[23,25,26]. Moreover, conventional nanomaterials coating techniques, such as dip coating or spray coating, are not suitable for micro-scale LMPs[22,27]. Previously, the integration of LM or LMP into fiber involves encapsulating them with elastomers[28,29]. However, this approach limits the ease of integration with electronic components and restricts their use as direct electrodes due to the presence of the encapsulation layer. Additionally, the inherent instability of LM or LMP poses a significant challenge, primarily because of the potential for leakage.

In this study, we introduce a solution process for coating fibers with a bi-layered structure of highly durable, stretchable, and initially conductive LMPs, which we refer to as BiLMPs as illustrated in Fig. 1a. Specifically, a suspension shearing-based coating process is employed, which involves two distinct stages. The first stage entails the coating of the fiber with polymer-attached LMPs (PaLMPs) to promote stretchability. Subsequently, the second stage involves the coating of the top layer with CNT-attached LMPs (CaLMPs) to improve mechanical durability and impart initial conductivity. The overall process of BiLMP coating on the fiber is presented in Supplementary Fig. 1.

Unlike solid conductive fillers, the semi-solid conductive filler in our BiLMPs remains stable under strain, resulting in the lowest gauge factor among conductive filler-coated fibers, as presented in Fig. 1b (See Supplementary Fig. 2 for the references)[30–36]. As a result, our BiLMP-coated fiber can be used as a stretchable interconnect, as shown in Fig. 1c. Additionally, the enhanced mechanical durability of our BiLMP-coated fiber ensures that there is no leakage or rupture under the application of mechanical stress such as rubbing (Fig. 1d). Given its remarkable electrical and mechanical properties, coupled with its capacity for high structural versatility (i.e., insert-ability, suture-ability, wind-ability, and knot-ability), and favorable biocompatibility, our BiLMP-coated fiber is highly suitable for use in minimally invasive 1D bioelectronics, such as brain activity monitoring electrodes, optogenetic neural probes, and peripheral nerve stimulators as portrayed in Fig. 1e.

## Results

### Ink preparation and coating process of deformable conductive filler

In order to produce a solution-processable ink that contains stable and stretchable LMPs, a suspension was formulated with tip sonication. This ink contains polymer-attached LMPs, referred to as PaLMPs, which are dispersed in a diluted weak acid solution (acetic acid in deionized water). The preparation process for the ink is illustrated in Supplementary Fig. 3, while the rheological properties of the ink are provided in Supplementary Fig. 4.

The typical method for coating conductive fillers on fibers, such as silver nanowires and CNTs, is dip-coating[2,37,38]. However, this method is generally accepted for depositing nano-scale fillers due to low capillary number of colloidal solution or suspension, resulting in very thin film[39,40]. Given that LMPs are typically micro-scale, the thin coating of the solution during dip-coating is inadequate to transport particles to the fiber, since it is not capable of overcoming gravitational force, as illustrated in Fig. 2a[39–41]. Furthermore, relying solely on the electrostatic interaction between particles and the substrate is inadequate for achieving robust adhesion of such micro-scale fillers[18].

To address this issue, we employed suspension shearing, a meniscus-guided coating technique, to achieve a compact assembly and robust adhesion of PaLMPs on fibers (the scaling and rheological analyses of dip coating and shearing method are further discussed in Supplementary Fig. 5)[40,42]. The meniscus is defined as the curved interface between air and solution that is naturally formed during the dragging of pinned solution on the surface. The high surface-to-volume ratio near the meniscus, coupled with the heated substrate, accelerates the evaporation of the solvent[42,43]. As illustrated in Fig. 2b, the suspension shearing-based coating can be divided into three distinct regions. In region ①, particles are transported towards the meniscus by capillary flow, Marangoni flow, and drag (boundary-driven) flow[42,43]. The continuous supply of particles is crucial for achieving a compact assembly of particles with a small capillary number coating flow (Supplementary Fig. 6). Other coating methods, such as doctor blading and spin coating, which do not provide a continuous supply of particles, are not suitable for compact and uniform particle assembly (Supplementary Fig. 7).

The solvent of our ink contains a diluted acetic acid, namely AA. Due to the negative charge of the PaLMPs, protons from AA can neutralize the negatively charged surface, thereby reducing the electrostatic repulsion between the particles (Supplementary Fig. 8)[44]. During suspension shearing, the concentration of protons increases as DI evaporates more rapidly than AA (Supplementary Fig. 9, which shows the pH value according to evaporation), further reducing repulsion between the negatively charged PaLMPs. This facilitates the compact deposition of particles on the fiber, as illustrated in region ②.

The achievement of robust particle adhesion to a substrate that endures mechanical deformation is imperative. To this end, chemical annealing of the particles in proximity to the substrate surface through the application of elevated temperature and low pH, engenders a partial removal of the oxide layer, and the extrusion of LM[18]. Reformation of the oxide layer with the extruded LM near the surface allows the robust adhesion between particles and substrate as illustrated in region ③ (see Supplementary Fig. 10 for robust adhesion of the bottommost layer)[45,46]. In contrast, when the chemical annealing process is not performed, the PaLMP film remains covered with a polymer layer. As a result, the desired tight adhesion is not achieved, leading to the easy delamination of particles, as demonstrated in Supplementary Fig. 11. Additionally, if the substrate is not subjected to heating, the chemical annealing of particles necessary for robust adhesion is not fully realized.

The effect of different ink combinations on the film coverage ratio is demonstrated in Fig. 2c, supporting the analysis presented above. Achieving enhanced wettability of the suspension and mitigating the repulsion between particles are critical factors for achieving a compact coating. Without the PSS acting as surfactant, the ink wettability is inadequate (Supplementary Fig. 12), resulting in partial coating of the substrate. Conversely, in the absence of AA, negatively charged particles repel each other during suspension shearing, impeding compact coating on the substrate. Only an acidic ink with enhanced wettability achieves a compact and uniform coating all over the substrate, including the textured substrate.

Conventional solid-filler coating strategies presented in Fig. 2d are not sufficient to compactly coat LMP on fibers. Dip coating is inadequate to create a conductive pathway as the particles are not compactly or robustly deposited on the fiber. Soaking also fails to achieve the desired result, as prolonged annealing in a heated AA leads to the formation of gallium oxide, which is neither conductive nor stretchable[47]. In contrast, semi-solid PaLMP is compactly coated into the fiber matrix using suspension shearing (Fig. 2e) owing to the

unique combination of evaporation-induced close-packed assembly and chemical annealing. Finally, an iterative shearing process was performed with CaLMP ink on the PaLMP-coated fiber to achieve BiLMP as presented in Supplementary Fig. 1. SEM image of coated BiLMP film is presented in Supplementary Fig. 13.

## Electrical and mechanical property of BiLMP

One major obstacle in the application of LMPs is the need for an electrical activation step, as the native oxide layer formed on the surface of LMPs acts as an insulator. Typically, mechanical or chemical activation, such as stretching/scrubbing or chemical etching, is

employed to remove the surface oxide layer and attain electrical conductivity[24,48,49]. However, such techniques often lead to rupturing of particles during electrical activation, reintroducing the problem of bulk LM (leakage, instability, and high surface tension). Our BiLMP, consisting of PaLMP and CaLMP layers, overcomes these challenges, as it exhibits high electrical conductivity while maintaining a bi-layer packed morphology without leakage on the surface (Supplementary Fig. 13). CaLMP contains CNTs, which form a network for electrical conductivity and mechanical durability, as reported previously, while PaLMP is partially annealed by acetic acid during the shearing process. As presented in Fig. 3a, although both films independently do not

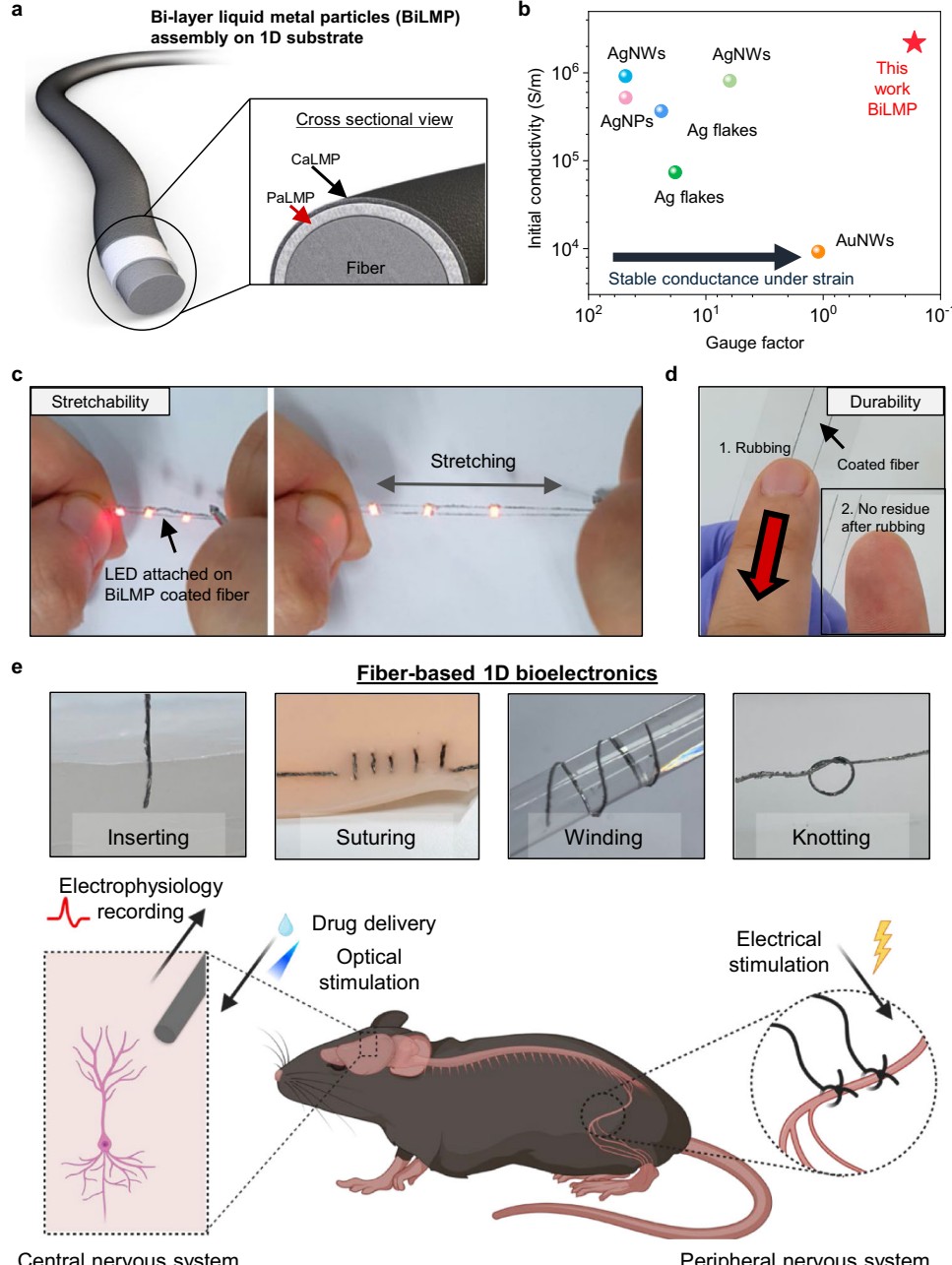

**Fig. 1 | BiLMP-coated fiber. a** Schematic illustration of BiLMP-coated fiber. The BiLMP-coated fiber consists of two layers: PaLMP (polymer-attached LMP) and CaLMP (carbon nanotube-attached LMP) **b** Initial conductivity and gauge factor of the BiLMP-coated fibers in comparison to previously reported conductive filler-based fibers. **c** Images demonstrating the stretchable operation of LEDs attached to the BiLMP-coated fiber before (left) and during (right) stretching. **d** Images

demonstrating the durability of BiLMP-coated fibers upon scrubbing. **e** Images of various mechanical interactions enabled with BiLMP-coated fibers for 1D bioelectronics (top) and schematic illustration (bottom) demonstrating recording and stimulation interfaces for implantable bioelectronics. Figure 1e was created with BioRender.com.

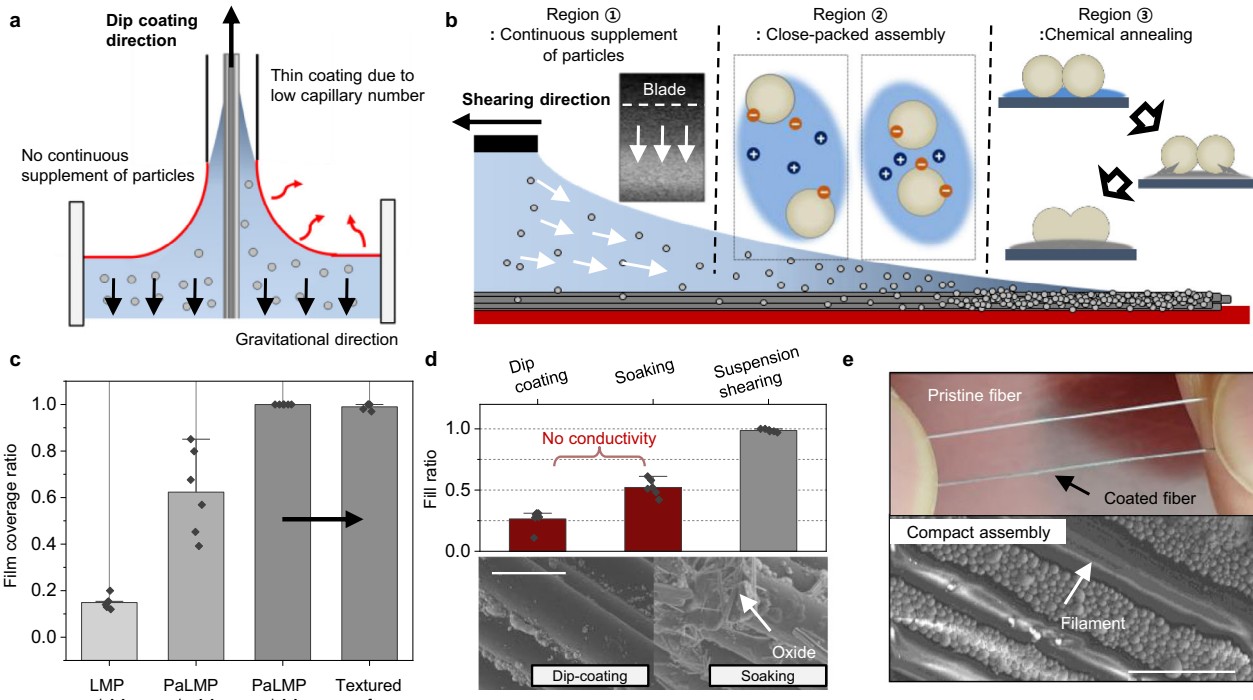

**Fig. 2 | Deposition of particle-assembled LMP on 1D substrate with suspension shearing. a** Schematic illustration of the incompatibility of LMPs with the dip-coating on fiber. **b** Schematic illustration of suspension shearing-based coating of LMP on fiber. **c** Film coverage ratio according to inks with different combinations of additives with suspension shearing. **d**, Comparison of fill ratio according to the coating technique. Scale bar: 50 μm. **e** Photograph and SEM image of PaLMP-coated fiber through suspension shearing. Reproducibility: Suspension shearing was conducted six times, and on each occasion, it resulted in a compact coating. Scale bar: 40 μm. Values in **c–d** represent the mean and the 1.5 IQR (*n* = 6).

exhibit sufficient conductivity for use in electrical devices, when CaLMP is coated on top of the PaLMP layer, the resulting bi-layered structure results high conductivity (2.24 × 10⁶ S/m, Supplementary Fig. 14 show the conductance of different type of fibers). The reason for such high conductivity can be attributed to the fact that CaLMP is in a dimethyl sulfoxide (DMSO)-based solution (Detail information regarding ink for solution process is presented in Supplementary Figs. 15 and 16.), and when PaLMP is exposed to DMSO, the detachment of PSS polymer from LMP and consequent cohesion of particles is likely to be induced (Supplementary Figs. 17 and 18). This effect is similar to that of the phase separation of PEDOT:PSS under DMSO exposure, which has been previously studied[50,51].

The stable conductance against physical deformation is a desirable characteristic for stretchable electronic. However, traditional stretchable conductors that use rigid conductive fillers such as metal nanowires and nanoparticles suffer from a significant reduction in conductivity with strain due to the percolation pathway being disrupted[13,15]. In contrast, our deformable conductive filler, known as LMP, exhibits 'positive piezo conductivity', which compensates for the geometrical elongation by increasing the contact area between particles, as explained further in Supplementary Fig. 19[18,21,52]. As a result, our PaLMP and BiLMP film shows almost constant resistance under the application of strain as shown in Fig. 3b (Further discussion regarding resistance change of BiLMP film is presented in Supplementary Fig. 20). On the contrary, CaLMP-coated conductive fiber undergoes an increase of resistance which is mainly attributed to the presence of rigid CNT percolation disrupted with strain. The reliable operation capacity of BiLMP-coated fiber under repeated strain is demonstrated by the cycle test (Fig. 3c). Additionally, the long-term stability and temperature stability of the BiLMP-coated fiber are presented in Supplementary Figs. 21 and 22.

The mechanical durability of the conductor is crucial for integration with other electronic components and application in bio-interfaced electrodes[19,53]. However, in the case of bulk LM, its fluidity causes instability and hinders its application as electrodes[25]. Although, PaLMP shows enhanced stability by the encapsulation of polymer on the surface of the particles; this polymer-oxide skin is easily ruptured and peeled off as presented in Fig. 3d. Conversely, the incorporation of CNTs in CaLMP and BiLMP has resulted in significant improvements in mechanical durability. Both films exhibit no detachment after peel-off testing with scotch tape unlike PaLMP film (Supplementary Fig. 23) and no rupturing of particles after scratching (Supplementary Fig. 24). The enhanced mechanical durability of BiLMP film enables the functionalization of electrodes like conventional metal (see Supplementary Fig. 25 for functionalized BiLMP-based bioelectrodes.).

In addition, Fig. 3e shows the mechanical robustness of fibers coated with BiLMP through repeated peel-off tests. The coated material exhibited no ruptures or detachment, while maintaining its initial electrical conductivity. This mechanical durability enables the integration of BiLMP-coated fibers with conventional electronics, such as LEDs or resistors (Supplementary Fig. 26), which is infeasible with unstable bulk LM coatings. The integration of BiLMP-coated fibers with conductance-stable interconnects allows electronic components to operate stably under strain (Fig. 3f), as well as the demonstration of sewn electrical circuits (Fig. 3g) without any leakage.

The mechanical robustness and large-scale coating capability of BiLMP enable the integration of conductive fibers onto commercial cloth using a sewing machine, as demonstrated in Fig. 3h. Furthermore, the large-scale smart clothes with electronic components (Fig. 3i) serves as further validation of the practicality of BiLMP-coated fibers, which show stable operation even under mechanical deformation. Taken together, these demonstrations highlight the versatility and usefulness of BiLMP-coated fibers in wearable applications.

## BiLMP-coated 1D bioelectronics
Due to the adaptability of meniscus-guided coating, BiLMP film can be coated on various types of fibers such as stretchable multifilament

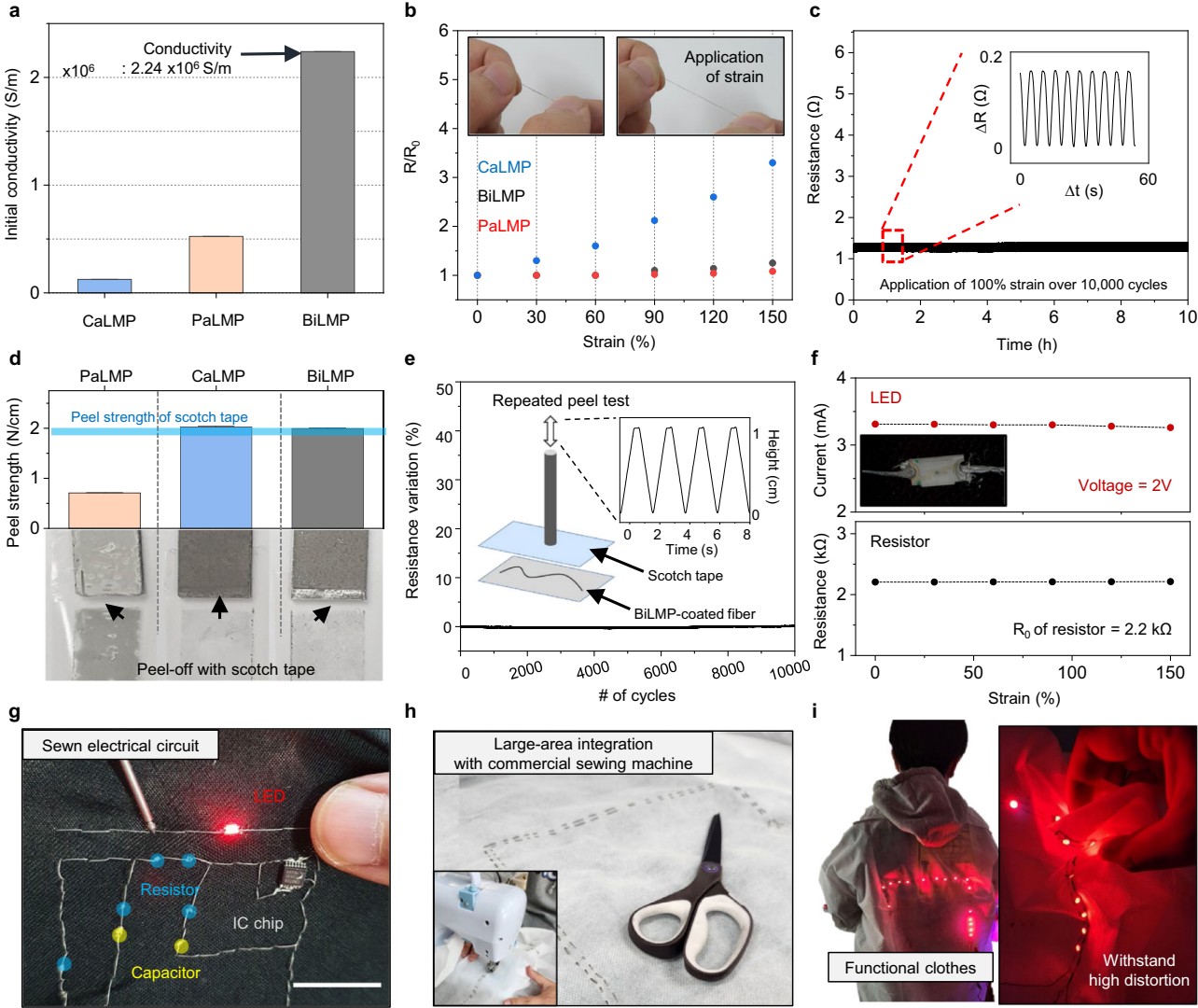

**Fig. 3 | Stretchability and mechanical durability of BiLMP-coated fibers. a** Initial electrical conductivity comparison of CaLMP, PaLMP, and BiLMP. **b**, Relative resistance of conductive fibers under strain with respect to the type of coated film (CaLMP, PaLMP, and BiLMP). **c** Relative resistance of BiLMP-coated fibers under 10,000 strain cycles. **d** Peel test of each film with commercialized scotch tape. There is no notable peel-off and residue with CaLMP and BaLMP films. **e** Resistance variation of BiLMP-coated fibers under repeated attachment and detachment of tape. **f** Electrical current passing through an attached LED and the resistance of the connected BiLMP-coated fiber interconnect under strain. **g** Photograph of a sewn electrical circuit with a BiLMP-coated fiber. Scale bar: 2 cm. **h** Image of large-area integration of BiLMP-coated fibers on commercial cloth. **i** Image of BiLMP fiber-integrated smart clothes.

fiber (polyurethane (PU) fiber), commercial threads, and thermally drawn multifunctional polymer fiber, and soft threads (Stiffness of each fiber type is presented in Supplementary Fig. 27). The inserting, wrapping, and knotting capabilities of BiLMP-coated fibers facilitate intimate interaction with various biological tissues, such as the central and peripheral nervous systems.

Fiber-type brain implants enable the recording of intracortical potentials with high spatiotemporal resolution[7]. We demonstrate the application of stretchable BiLMP-coated recording fibers (BiLMP-R-fibers) for electrophysiology in the mouse brain hippocampus CA1 region through successful recording of endogenous neural activity even in stretched conditions (Fig. 4a–c). The extracted single units recorded from BiLMP-R-fibers in both pristine (0% strain) and stretched (20% strain) states and analyzed through principal component analysis (PCA) demonstrate that neuronal activity is recorded from the same neuronal population, thus indicating that signal quality will be maintained under mechanical deformations (Fig. 4d, e).

The versatility of BiLMP meniscus-guided coatings further enable applications for flexible, multifunctional fiber devices fabricated with the thermal drawing process (TDP) (Supplementary Fig. 28) for simultaneous optogenetic stimulation, electrophysiological recording, and drug delivery (Fig. 4f)[4,5]. Multifunctional BiLMP fibers (BiLMP-M-fibers) successfully recorded optically-evoked potentials from CA1 regions of *Thy1:ChR2-YFP* mice during 10 Hz optical stimulation (Fig. 4g). The decayed response exhibited during 100 Hz stimulation suggests that the evoked neural activity is not a consequence of light-induced artifacts (Supplementary Fig. 29). Additionally, the AMPA receptor antagonist, CNQX, is injected through the microfluidic channel of the BiLMP-M-fiber with simultaneous optical stimulation and electrophysiology. The peak-to-peak amplitude of evoked signals is dramatically diminished after CNQX injection (Before: $599.7 \pm 147.9\mu V$, after: $234.3 \pm 69.8\mu V$). This result confirms the robust integration of BiLMP-coatings with multifunctional fibers for simultaneous drug delivery, optical stimulation, and electrophysiological recording (Fig. 4h, Supplementary Fig. 30).

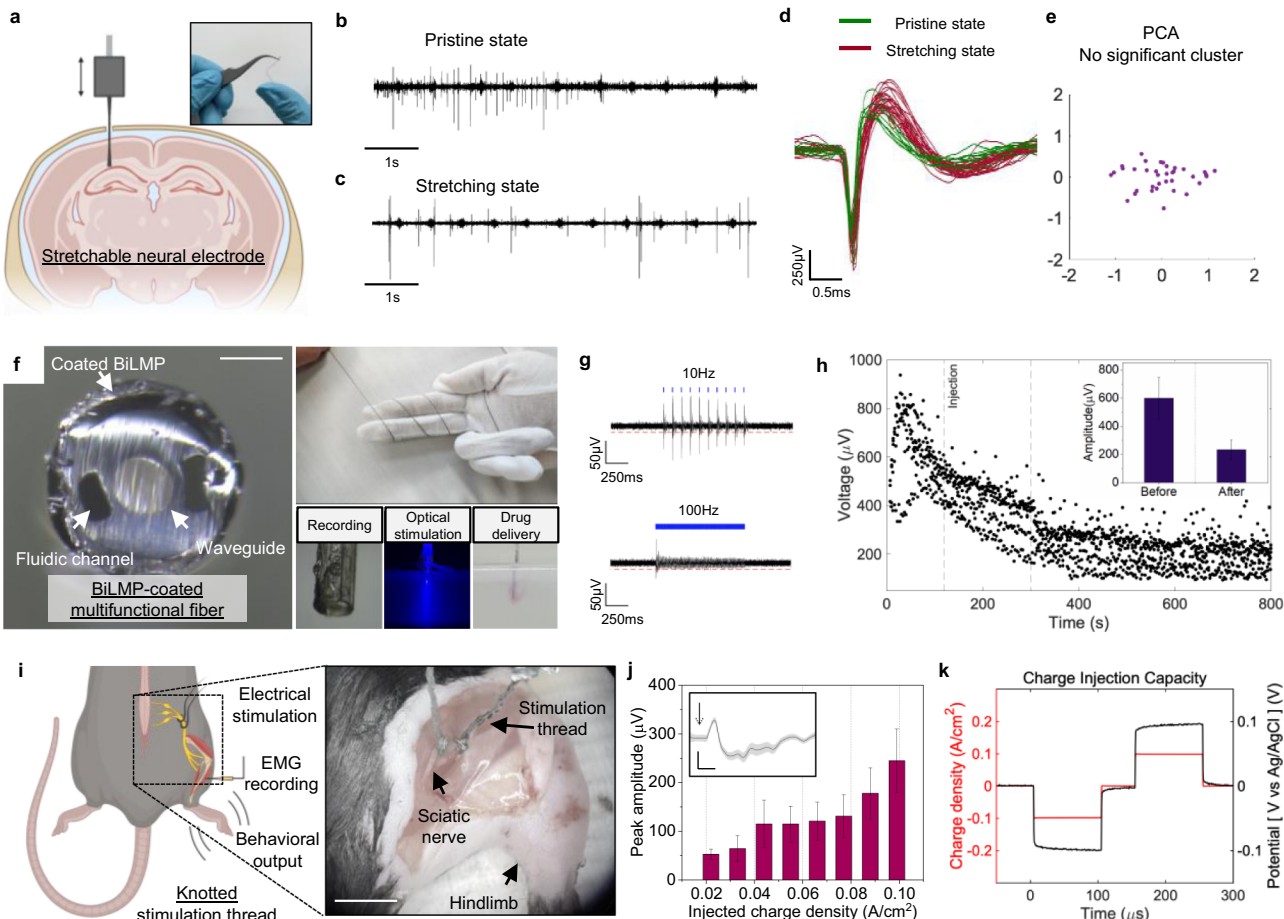

**Fig. 4 | Neural recording and stimulation with BiLMP-coated fibers for multi-functional 1D bioelectronics. a** Image and illustration of BiLMP-coated recording fiber (BiLMP-R-fiber) for electrophysiological recording. **b** Electrophysiology recordings of spontaneous activity from CA1 regions with BiLMP-R-fiber before stretching. **c** Electrophysiology recordings of spontaneous activity from CA1 regions with BiLMP-R-fiber after stretching (20% strain). **d** Overlapped spike waveforms recorded from the BiLMP-R-fiber before (green) and after (red, 20% strain) stretching. **e** PCA clustering of spikes recorded from BiLMP-R-fiber before and after stretching. **f** Cross-sectional image of BiLMP-multifunctional (extracellular recording, optical stimulation, and drug delivery) fiber device (BiLMP-M-fibers). Reproducibility: Suspension shearing was conducted six times, and on each occasion, it resulted in a compact coating. Scale bar represent 100 μm. **g** Electrophysiology recordings of optically evoked potentials (OEP) with 10 Hz (top), 100 Hz (bottom) optical simulation, simultaneously recorded through

BiLMP-M-fibers. **h** Inhibition of OEP peak potentials with time, after injection of synaptic blocker (CNQX) using BiLMP-M-fibers. Inset: peak-to-peak amplitude of OEP at before/after CNQX injection. Data are presented as mean values +/- SD. Each 180 peaks are used in sampling. **i** Illustration (left) and image (right) of sciatic nerve electrical stimulation using BiLMP-based stimulation thread devices (BiLMP-S-threads) with simultaneous EMG recording. Scale bar represent 10 mm. **j** Correlation between peak amplitude of evoked EMG signal and injected charge density from the BiLMP-S-thread. Data are presented as mean values +/- SD. Each 50 peaks are used in sampling. Inset: raw data of EMG signal. Error band is presented as mean values +/- SD. Scale bar represent 300μV (vertical) and 0.01 s (horizontal). The arrow indicates the onset of a biphasic electrical stimulation pulse of 0.1 A/cm². **k** Recorded charge injection capacity waveform of the BiLMP-S-thread. Figure 4a, i were created with BioRender.com.

Lastly, we demonstrate the neuromodulation capability of BiLMP-based stimulation thread devices (BiLMP-S-thread). The BiLMP-S-thread (200μm in diameter) is tied to the sciatic nerve of wild-type mice. Upon application of electrical stimulation, hindlimb movement was observed in conjunction with electromyography (EMG) signals in the tibialis anterior muscle (Fig. 4i). Upon application of 10 Hz and 100 Hz electrical stimulation, the hindlimb demonstrated a periodic tremble and loss of synchrony due to its muscular physiology. Furthermore, the recorded EMG signal increased proportionally with injected charge density, demonstrating successful downstream stimulation of the leg muscle (Fig. 4j). To identify the stimulation safety limits of the BiLMP-S-thread, the charge injection capacity (CIC) was measured. Electrode polarization was measured during 0.1 A/cm² biphasic current injection (voltage drop: −0.1 V) to verify that polarization potential is below the hydrolysis voltage, demonstrating that the BiLMP-S-thread can be utilized for stimulation without inducing undesirable biochemical reactions (Fig. 4k, Supplementary Fig. 31)[54].

Finally, in vivo biocompatibility and long-term operation of BiLMP-based bioelectronics were demonstrated to ensure its reliability as implantable electrode (Supplementary Figs. 32 and 33).

## Discussion

Integration of deformable and stretchable conductive filler on fiber with high durability and low gauge factor can open new opportunities for wearable and implantable electronics by providing stable operation capability under mechanical deformation. Here, we demonstrate an ink preparation and coating technique for the realization of LMP-based stretchable conducting fibers. Suspension shearing of PaLMP ink allows the compact assembly of conductive filler on stretchable and functionalized fiber. By subsequent coating of durable CaLMP dispersed in DMSO, initially conductive and mechanically durable BiLMP-coated fibers are fabricated. These properties are realized through the interaction between polymer and polar-organic solvent and the post-annealing process. BiLMP shows stable conductance

under strain unlike conventional rigid metal fillers, which realize stretchable conducting fiber with the lowest strain gauge among conductive filler-coated fiber while showing metal-level electrical conductivity. Demonstrations of sewn electrical circuit, smart clothes, stretchable 1D bioelectrode, and multifunctional optical probes validate the effectiveness of BiLMP as an electrical filler in fiber electronics.

## Methods

### Experiments on human and animal subjects

All experiments on wearable and implantable applications were performed under approval from the Institutional Review Board at Korea Advanced Institute of Science and Technology (protocol number: KH2021-039). All human subjects voluntarily participated in the experiments after providing informed consent.

### Materials

Unless otherwise stated, all the chemicals used in the current work were used without further purification. Eutectic gallium indium liquid metal (EGaIn) was purchased from Changsha Ruichi Nonferrous Metals. Poly(styrene sulfonate) (PSS, M.W.~1,000,000), and acetic acid (99%), dimethyl sulfoxide (DMSO) (99.5%), nitric acid (70%), and acetic acid (99%), $Pt(NH_3)_4(NO_3)_2$ (Pt precursor) were purchased from Sigma-Aldrich. Single-wall CNTs was purchased from OSSiAl. Commercial Kevlar fiber, commercial sewing thread, polyurethane fiber, and fabricated multifunctional fiber were employed.

### Preparation of PaLMP ink

The synthesis of PaLMP ink involved the combination of 2 g of EGaIn, 0.14 g ($1.4 \times 10^{-7}$ mol) of PSS, and 2 mL of diluted acetic acid (5 vol.% in deionized water) in a 10 mL vial. The mixture was subjected to a probe sonication (VC-505, Sonics & Materials, 3 mm microtip) at a 30% amplitude for a duration of 30 minutes. Following the tip sonication step, additional acetic acid (10 vol.%) was added to facilitate chemical sintering.

### Functionalization of SWCNTs for CaLMP ink

The carboxylation of CNTs involved immersing them in 70% nitric acid at 90 °C for a duration of 4 hours. The resulting carboxylated CNTs were collected on a 0.22 μm pore size membrane filter (Millipore) through vacuum filtration, followed by rinsing with DI. To functionalize the carboxyl groups of the CNTs with Pt precursors, a 5 wt% Pt(NH3)4(NO3)2 solution was dispersed in the aqueous solution of carboxylated CNTs at 40 °C for a duration of 1 hour. Subsequently, the Pt precursor-decorated CNTs were collected through vacuum filtration and dried in an oven at 80 °C for 1 hour. Finally, Pt nanoparticles were formed on the surface of the CNTs by reducing them at 300 °C in a hydrogen ($H_2$) atmosphere for a duration of 2 hours.

### Preparation of CaLMP ink

The preparation of CaLMP ink involved collecting 2 g of EGaIn, 0.006 g of Pt-decorated carbon nanotubes (Pt-CNTs) that were previously synthesized, and 6 mL of diluted acetic acid (5 vol% in DMSO) into a 20 mL vial. Subsequently, a probe sonication (VC-505, Sonics & Materials, 12.7 mm solid tip) was applied at a 70% amplitude for a duration of 20 minutes.

### PaLMP coating on fiber via solution shearing

A customized blade coating machine was employed to coat PaLMP films onto a fiber substrate. In order to conduct desired solution shearing, the blade was tilted to 5° and the gap between the blade and substrate was maintained at 200 μm. Prior to the solution shearing process, the fibers were attached to the glass substrate using polyimide tape and subjected to treatment with oxygen plasma (Femto Science) at 100 W for 1 minute, in order to clean and enhance wettability of the surface. To ensure reliable coating and achieve electrical

conductivity, we optimized the coating conditions. The process involved heating the glass substrates to 70 °C. We introduced 200 μl of PaLMP ink between the blade and the substrate, which was then coated on the substrate at a rate of 0.7 mm/s. This controlled temperate and speed allowed for the efficient evaporation of the solvent during the shearing process. Following the shearing process, the PaLMP-coated fiber underwent further heating on a hot plate set at 70 °C to ensure complete evaporation of any residual solvent.

### Fabrication of BiLMP-coated fiber

The fabrication process for the BiLMP layer incorporated an additional shearing coating step utilizing CaLMP ink applied to PaLMP-coated fiber. This process was conducted at a temperature of 80 °C and a speed of 2 mm/s. Subsequently, the coated fiber was placed on a hot plate at 80 °C until all residual solvents evaporated.

### Fabrication of multifunctional fiber through thermal drawing process

Before scaling down the process through custom-made tower, the macroscopic 'Preform' was prepared with the desired structure of the microscopic fiber. The preform was composed of transparent thermoplastic polymer polycarbonate (PC; Goodfellow) and polymethyl methacrylate (PMMA; Goodfellow), which had similar viscosity, glass transition temperature, and melting temperature for stable TDP. To form PC core and PMMA cladding, 6.35 mm diameter PC rod were wrapped in PMMA film and consolidated at 180 °C. The 16 mm diameter preform was machined two 3.8 mm grooves with CNC machine. And then, additional PC film was wrapped for substantial drawing. The final preform (19 mm diameter) was heated and drawn at 150 - 170 °C with furnace and capstan in our tower system. Through TDP, 200 microns of multifunctional fiber with same cross section was manufactured (95 times scale down). For our experiment, the microscale fiber, which has the capability of optical stimulation, drug delivery, and electrophysiological recording, was used.

### In vivo monitoring applications

All application procedures on the mouse model were performed with approval by the Institutional Animal Care and Use Committee (IACUC) of KAIST. All experiments were carried out using the *thy1* promotor-Channel Rhodopsin 2 transgenic mouse line (*Thy1::ChR2* mice). For the recording of the ECoG signal, the surgical procedure of craniectomy is preceded. Mice were anesthetized with 4% isoflurane (flow rate: 1.0 L/min) induction and 1.5% isoflurane maintenance. The cranial window of a 2 mm x 2 mm scale was generated in a mouse skull with the surgical dental drill (centered on coordination AP: 2 mm, ML: 2 mm), and the recording was conducted with a filtering frequency range of 3 - 300 Hz and sampling rate of 3052 Hz. All neural signals were recorded using a LABRAT instrument, Synapse software, ZIF-Clip, and ZCA-MIL16 (Tucker-Davis Technologies). For the recording of single neuron recording with 1D bioelectronics, the hole for implantation was created by the surgical dental drill. To insert fiber without buckling effect, we coated BiLMP-R-fibers with polyethylene glycol (MW 6000, Sigma Aldrich). The devices were inserted on a stereotaxic frame (Digital Stereotaxic Instrument 68025, RWD Life Science Corp., Shenzhen, China). The extracellular electrophysiological recording was conducted with a filtering frequency of 300 - 5000 Hz and the sampling rate of 24414 Hz. For chronic implantation, dental cement (Lang, Ortho-jet) was used for protecting the device and mounting. For the optogenetic experiment, optical stimulation is applied with the blue light laser (RWD, LIOG473-80-A1) and a power meter (Thorlabs, S121C and PM100D) was used as the sensor to check the output light intensity. The optical stimulation is 10 mW at the tip and 5 ms pulse-width. In the experiment to verify the stable electrophysiological recording during stretching deformation, we fixed the fibers to the mouse cranium with dental cement, and upper part of them were elongated (20%

strain, initial length 5 mm) with a three-dimensional manipulator of stereotaxic instruments. To avoid undesired artifacts, all recording instruments have been fixed except for fiber itself.

## In vivo electrical stimulation application

The experimental procedures were performed according to the abovementioned conditions. For verifying the stimulation capability of BiLMP-based thread, C57BL/6NHsd mice was conducted skin incision on the dorsal aspect of the hindlimb muscle. After identifying biceps femoris muscles and surrounding tissue, flexible threads were knotted on sciatic nerve. Through an isolated pulse stimulator (MODEL 2100, A-M Systems, Carlsborg, WA, USA), electrical stimulation was induced. The EMG signals were recorded by LABRAT instrument.

## Immunohistochemistry

For assessing foreign body response, BiLMP-based 2D substrate was implanted in 17 male C57BL/6NHsd mice aged 6–8 weeks (KOATECH, Gyunggi-do, South Korea) after cranial window craniectomy. The group of 2 weeks ($n = 8$) and 2 months ($n = 9$) after implantation were perfused with 4% paraformaldehyde (PFA) in phosphate buffer solution (PBS). After frozen in OCT compound (Tissue-Tek 4583; Sakura Finetek Inc, Torrance, CA, USA), the 40µm thickness of the coronal section (AP: −2mm) was obtained by cryostat (Leica Microsystems, Wetzlar, Germany). The sections were dyed with goat anti-GFAP primary antibody (1:1000, ab53554, Abcam), donkey anti-goat labeled with Alexa Fluor 488 (1:1000, A11055, Invitrogen) for GFAP marking and rabbit anti-CD68 primary antibody (1:500, ab125212, Abcam), donkey anti-rabbit labeled with Alexa Fluor 594 (1:1000, A21207, Invitrogen) for CD68 marking. The mounting onto microscopic slides was conducted with the mounting medium with DAPI, 4′,6-diamidino-2-phenylindole (Vectashield, Vector Laboratories, Burlingame, CA, USA). A laser-scanning confocal microscope (C2; Nikon, Tokyo, Japan) was used for imaging the brain section and ImageJ software was used for image analysis.

## Characterization

In order to examine the morphology of the coated PaLMP, CaLMP, and BiLMP, SEM images were captured using an S4800 scanning electron microscope (Hitachi). Real-time monitoring of EMG was carried out using a commercial wireless electrophysiology measurement equipment (BioRadio, Great Lakes NeuroTechnologies).

For electrical characterization, a LCR meter (4284 A, HP) was employed to measure the electrical properties of the conductive fiber. To monitor resistance variation under strain, a force gauge (with a maximum force of 50 N, Mark-10), a stand with a motor (Mark-10), and a customized manual strain machine were utilized. To eliminate any potential geometrical factors, the conductivity of each layer is measured by coating the films on glass substrates.

Chemical characterization involved measuring the zeta potential values of the inks using dynamic light scattering (Zetasizer nano zs, Malvern). Each ink was characterized using a UV/VIS spectrophotometer (Lambda 1050, Perkin Elmer) under wavelengths ranging from 200 nm to 500 nm.

The apparent viscosity of the LM inks was determined using an MCR 302 rheometer (Anton Paar) at a shear rate of $10^{-2} - 10^{2}\,s^{-1}$ at room temperature. To evaluate the wettability of each ink, the contact angle was measured twice for each sample: once when 100 µl of the sample was dropped, and once when 50 µl was withdrawn, using a contact angle analyzer (SEO Phoenix).

## Reporting summary

Further information on research design is available in the Nature Portfolio Reporting Summary linked to this article.

## Data availability

The authors declare that the data supporting the findings of this study are available within the article and its Supplementary Information files. Extra data or source files are available from the corresponding author.

## Code availability

The software code used for analyzing neural signal is available from the corresponding author.

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

## Acknowledgements

This work was supported by the National Research Foundation of Korea (NRF- 2022R1A2C2006076, 2021R1A2C4001483, RS-2023-00207970, and 2022M3H4A1A03085346) and KAIST Global Singularity Research Program, KAI-NEET Seed Money Project and Post-AI Project. G.-H.L. has been supported by the POSCO Science Fellowship of POSCO TJ Park Foundation.

## Author contributions

G.-H.L., D.H.L., Woojin J., Seongjun P., J.-W.J., and Steve P. conceived the concept and designed experiments. G.-H.L. designed the ink, performed chemical and electrical characterization, and conducted data analysis. D.H.L. conducted the experimental work and performed characterization. Woojin J. conducted the animal experiment. J.Y, K.A, and J.N. conducted an analysis regarding the rheological behavior of ink. J.K.K. and W.J. assisted the fabrication of functionalized CNT. Y.H.K., J.J. characterized the fibers. M.K., K.S.N. and J.L. provided comments regarding the manuscript and data analysis. Seongjun P., J.-W.J., and Steve P. were responsible for managing all aspects of this project. G.-H.L., D.H.L., and Woojin J. wrote the draft. Seongjun P., J.-W.J., and Steve P. revised the manuscript. All authors discussed the results and the manuscript.

## Competing interests

The authors declare no competing interests.
