## [Peer Review File · Nature Communications]

REVIEWER COMMENTS

Reviewer #1 (Remarks to the Author):

In this study, the authors have presented a novel conductance-stable stretchable fiber, achieved through the integration of LMP on the fiber. The resulting fiber electrode exhibits stable conductivity, mechanical durability, and biocompatibility - all of which are crucial factors for the development of fiber-based electrodes. Notably, the gauge factor of this conductive filler-coated fiber is the lowest reported thus far.

The authors have successfully demonstrated the practical application of this fiber by constructing a sewn-electrical circuit on clothes and a 1D bioelectrode. The work introduces several innovative concepts, such as shearing-based coating on fiber and CNT-included bi-layer composite, which further enhance its scientific significance.

Given the above, this reviewer recommends accepting the manuscript for publication in Nature Communications. However, before publication, the authors are advised to revise the manuscript to enhance its overall clarity and scientific content.

1. The authors should add a clarification regarding the difference between the fiber electrodes demonstrated in this paper and conductive fibers that utilize liquid metal injection into hollow tubes.
2. The supplementary information includes rheological properties related to the capillary number. It would be helpful if the authors could discuss the rheological properties of the ink as the concentration of LMP is increased.
3. Is there any change in conductivity along with the diameter of coated fiber?
4. What is the minimum diameter of fiber shows the conductivity?
5. Is there a conductivity change along with time?
6. In the supplementary information, it would be beneficial to include a photograph of the coating process along with a schematic illustration.

Reviewer #2 (Remarks to the Author):

This manuscript "Conductance stable and mechanically durable bi-layer EGaIn composite-coated stretchable fiber for 1D bioelectronics" presents a solution method that robustly and compactly assembles mechanically durable and initially conductive liquid metal particles on fibers. The devices exhibit exceptional durability, electrical conductivity, stretchability, and biocompatibility. The authors also demonstrated applications of these 1D electronic devices in sewn electrical circuits, smart clothes, stretchable biointerfaced fiber, and multifunctional fiber probes. This work introduces an innovative method for producing mechanically durable and initially conductive coating on fibers which can enable a broad range of applications. However, there are several concerns (mostly minor) that need to be addressed before the publication of this manuscript.

1. There are many publications on using liquid metal electrodes in stretchable fibers, such as *Science advances*, 7(22), p.eabg4041, *Advanced Materials*, 30(27), p.1707251. Please clarify what the main advantages of this work are compared to the previous works.
2. There may be various parameters, such as the fiber shape, fiber size, material properties, coating speed that could affect the output of the coating process. It is recommended that the authors conduct a parameter analysis.
3. How will the operation environment affect the performance (e.g. temperature, humidity)?
4. Fig. 4a-c shows the electrophysiological recording through coated fibers when the fibers in pristine (0% strain) and stretched (20% strain). More details should be provided on how the fiber was stretched by 20% when it is implanted in the mouse brain.
5. When the soft and stretchable fiber was implanted into the brain, did the authors encounter any problems (difficulties) with insertion? What are the size and stiffness of the fiber probes?

6. Please clarify what the sampling rates are for the neural recording (such as in Fig. 4d and Supplementary Fig. 24d). Also, scale bars should be added in Fig. 4f and 4i.

7. In the biocompatibility test (supplementary Fig. 25), the authors claimed there is no immune response based on the comparison of the histology data with/without devices. It is hard to see the location of the devices in the images. Please circle or indicate the location of the devices. The image quality is also not high.

Reviewer #3 (Remarks to the Author):

This manuscript reports the shearing-based deposition of polymer-attached liquid metal particles (PaLMP) and CNT-attached LMP (CaLMP) to create bilayer LMP (BiLMP) on polymer fibers. The BiLMP-coated fibers exhibited the electrical conductivity (2.24×10^6 S/m), stretchability (150%), and biocompatibility. The BiLMP-coated fibers also exhibited stable conductance under strain with the lowest gauge factor among conductive filler-coated fibers in literature. The applications of BiLMP-coated fibers such as electrical circuit, smart clothes, and stretchable bioelectrodes are demonstrated. However, the chemical mechanism, oxidation study, and detailed electrical characterization are missing in the manuscript. The novelty in the fiber synthesis itself is doubtful although fancy demonstrations are carried out. The following concerns should be addressed properly before considering this work for publication in Nature Communications.

1. There is no explanation of why CNTs are functionalized with Pt nanoparticles prior to the LMP functionalization for the synthesis of CaLMP. What happens without the Pt coating? How is the LMP attached to CNT (chemical bonding mechanism)? What is the exact chemical bonding mechanism between PSS and LMP?

2. Fig. 3a. It is strange that the electrical conductivity of CaLMP is significantly smaller than that of PaLMP. Is this the conductivity of entire fiber or coating layer only? This should be clarified in the entire manuscript. Why is the conductivity of BiLMP higher since it is only the double layer of CaLMP and PaLMP? What is the thickness of each layer? What is the concentration of each component (e.g., CNT concentration in the entire composite fiber)? What is the concentration of EGaIn? What is the measured conductivity of EGaIn? Apparently, the CNT did not increase the conductivity. What is the major mechanism of achieving high conductivity of BiLMP-coated fiber? How did they measure the conductivity precisely?

3. Figure 1a schematic is confusing. How many BiLMP-coated fibers in the bundle? Each fiber has bi-layer coating? What fills the gap between fibers?

4. The authors claimed that the oxide layer of the LMPs was removed by acetic acid and reformed after coating, leading to better adhesion between particles and substrate. How did the authors confirm the removal and reformation mechanism of the oxide layer other than the schematic in Supplementary Fig.10. Any spectroscopic evidence?

5. Fig. 3d. The coating is only the physical adsorption. The scotch tape tests only the adhesion of the skin layer. The adhesion of PaLMP was bad. How stable is the adhesion of the bilayer on the polymer fiber?

6. The delamination of bare liquid metal particles under strain is provided in the supplementary Fig. 11. Do the BiLMPs delaminate under strain? This should be compared under the same condition.

7. In Fig. 3b, the authors explained the increase in resistance of the CaLMP-coated fiber is attributed to the disruption of rigid CNT percolation with strain. Then, why is there no such increase in resistance in BiLMP since its top layer is CaLMP? Provide the mathematical resistance modeling of this process.

8. In the electrical property explanation (line 177-185), the authors claimed that they overcome the oxide layer formation of LMPs by the synthesis of BiLMP. On the contrary, in line-142, they explained that the reformation of the oxide layer in the PaLMP enhances the adhesion to the substrate. The two claims are contradicting. The oxide formation and removal also need to be experimentally confirmed.

Response to Reviewers' Comments

The Reviewer's comments are in **bold** and revised texts are highlighted.

Reviewer #1

In this study, the authors have presented a novel conductance-stable stretchable fiber, achieved through the integration of LMP on the fiber. The resulting fiber electrode exhibits stable conductivity, mechanical durability, and biocompatibility - all of which are crucial factors for the development of fiber-based electrodes. Notably, the gauge factor of this conductive filler-coated fiber is the lowest reported thus far.

The authors have successfully demonstrated the practical application of this fiber by constructing a sewn-electrical circuit on clothes and a 1D bioelectrode. The work introduces several innovative concepts, such as shearing-based coating on fiber and CNT-included bi-layer composite, which further enhance its scientific significance.

Given the above, this reviewer recommends accepting the manuscript for publication in Nature Communications. However, before publication, the authors are advised to revise the manuscript to enhance its overall clarity and scientific content.

Answer. We appreciate the Reviewer for the helpful comment regarding our work. We have revised our work according to the Reviewer's comments. Point-by-point responses are attached below:

1. The authors should add a clarification regarding the difference between the fiber electrodes demonstrated in this paper and conductive fibers that utilize liquid metal injection into hollow tubes.

Answer. As pointed out by the Reviewer, previous methods of fabricating liquid metal-based fibers involved injecting the metal into a hollow fiber. However, this approach is not suitable for various bioelectronics applications due to the fact that the conducting electrode is encapsulated with elastomer. In contrast, in our case, the conducting electrode is implemented outside of the fiber, enabling easy integration with other electronic components.

Moreover, previous liquid metal fibers have encountered issues with leakage primarily caused by mechanical instability, which poses a significant obstacle to their integration with other electronic components. Conversely, our mechanically durable liquid metal composite ensures stable operation without any leakage.

In order to emphasize the novelty of our work, we have made revisions to the introduction section in Main Text as presented below:

Revised Main Text

Moreover, conventional nano materials coating techniques, such as dip coating or spray coating, are not suitable for micro-scale LMPs.^{22,27} **Previously, the integration of LM or LMP into fiber involves encapsulating them with elastomers.^{28,29} However, this approach limits the ease of integration with electronic components and restricts their use as direct electrodes due to the presence of the encapsulation layer. Additionally, the inherent instability of LM or LMP poses a significant challenge, primarily because of the potential for leakage.**

Added References

- 28 Zheng, L. J. *et al.* Conductance-stable liquid metal sheath-core microfibers for stretchy smart fabrics and self-powered sensing. *Science Advances* **7**, doi:10.1126/sciadv.abg4041 (2021).
- 29 Qu, Y. P. *et al.* Superelastic Multimaterial Electronic and Photonic Fibers and Devices via Thermal Drawing. *Advanced Materials* **30**, doi:10.1002/adma.201707251 (2018).

2. The supplementary information includes rheological properties related to the capillary number. It would be helpful if the authors could discuss the rheological properties of the ink as the concentration of LMP is increased.

Answer. As pointed out by the Reviewer, the meniscus-guided coating technique is influenced by various factors. One such factor is the content of the liquid metal precursor (LMP), which directly affects the viscosity of the ink and consequently impacts the capillary number during the coating process.

However, it is important to note that an increase in the insoluble contents can lead to a solution behavior like mud. This undesirable behavior can result in severe clustering and clogging, under high concentrations of LMP, which hinders the achievement of a compact coating on the fiber.

We have included the rheological properties of the ink with high concentration of LMP in the Supplementary Information.

Revised Supplementary Information

Supplementary Fig. 4| Viscosity curve of ink according to PaLMP concentration.

The addition of insoluble contents to the ink results in an increase in viscosity. At a concentration of 10 vol.%, the ink exhibits Newtonian behavior similar to that of bare solvent. However, at a concentration of 50 vol.%, the ink demonstrates non-Newtonian behavior with significantly increased viscosity. This high concentration of insoluble content leads to the formation of clusters, which is undesirable for solution shearing processes.

3. Is there any change in conductivity along with the diameter of coated fiber?

Answer. We appreciate the Reviewer's comment on the conductivity of the fiber. Our results align with the Reviewer's point, as we have also noted that an increase in the diameter of the coated fiber corresponds to an increase in the interconnect area. This, in turn, leads to an increase in the conductance of the fiber.

To provide further information, we have included a graph illustrating the relationship between the resistance of the fiber and its diameter.

Added Figure and caption in the Supplementary Information

Supplementary Fig. 14| Resistance of BiLMP-coated fibers. a, Resistance of BiLMP coated fiber according to type. **b,** Resistance of BiLMP coated rectangular shape fiber according to width of diameter.

To achieve reliable electrical conductivity, it is recommended to use fibers with a width of over 50 μm.

Revised Main Text

When CaLMP is coated on top of the PaLMP layer, the resulting bi-layered structure results high conductivity (2.24×10^6 S/m, **Supplementary Fig. 14** show the conductance of different type of fibers).

4. What is the minimum diameter of fiber shows the conductivity?

Answer. We thank the Reviewer for the helpful comment. We have experimentally found that 50 μm is the minimum diameter of a fiber showing electrical conductivity.

We have added a conductance of fiber according to the diameter of fiber in Supplementary Information.

Added Figure and caption in the Supplementary Information

Supplementary Fig. 14| Resistance of BiLMP-coated fibers. a, Resistance of BiLMP coated fiber according to type. **b,** Resistance of BiLMP coated rectangular shape fiber according to width.

To achieve reliable electrical conductivity, it is recommended to use fibers with a width of over 50 μm .

5. Is there a conductivity change along with time?

Answer. We thank the Reviewer for the helpful comment. In contrast to conductive polymers or silver nanowires that often experience a deterioration of electrical conductivity over time, our liquid metal composite exhibits stable conductance over an extended period. We have conducted experiments and found no significant changes in resistance during a 3-week timeframe

We have included the data regarding the stability of the conductance over time in the Supplementary Information.

Added Figure and caption in the Supplementary Information

Supplementary Fig. 21| Resistance variation of BiLMP film with time.

Revised Main Text

The reliable operation capacity of BiLMP coated fiber under repeated strain is demonstrated by the cycle test (Fig. 3c). Furthermore, the long-term stability of the BiLMP-coated fiber is presented in Supplementary Fig. 21.

6. In the supplementary information, it would be beneficial to include a photograph of the coating process along with a schematic illustration.

Answer. We appreciate the Reviewer for helpful review. We have added a photograph of coating process with a schematic illustration.

Revised Figure in the Supplementary Information

Supplementary Fig. 1| BiLMP coating on fiber. a, Schematic illustration of the BiLMP coating process on the fibers. **b,** Photograph of suspension shearing on fibers.

To fabricate the BiLMP-coated fiber, polyurethane fibers are fixed to the substrate. PaLMP film is coated with suspension shearing on the prepared substrate. Thereafter, CaLMP film is coated on the PaLMP film in the same way to make BiLMP-coated fiber.

Reviewer #2

This manuscript "Conductance stable and mechanically durable bi-layer EGaIn composite-coated stretchable fiber for 1D bioelectronics" presents a solution method that robustly and compactly assembles mechanically durable and initially conductive liquid metal particles on fibers. The devices exhibit exceptional durability, electrical conductivity, stretchability, and biocompatibility. The authors also demonstrated applications of these 1D electronic devices in sewn electrical circuits, smart clothes, stretchable biointerfaced fiber, and multifunctional fiber probes. This work introduces an innovative method for producing mechanically durable and initially conductive coating on fibers which can enable a broad range of applications. However, there are several concerns (mostly minor) that need to be addressed before the publication of this manuscript.

Answer. We appreciate the Reviewer for the acknowledgment of our work. We have revised our manuscript with the additional information to clarify the manuscript. Point-by-point responses are attached below:

1. There are many publications on using liquid metal electrodes in stretchable fibers, such as Science advances, 7(22), p.eabg4041, Advanced Materials, 30(27), p.1707251. Please clarify what the main advantages of this work are compared to the previous works.

Answer. We thank the Reviewer for the valuable comment. As highlighted by the Reviewer, there have been several endeavors to develop stretchable fibers using liquid metal. Traditional approaches for fabricating liquid metal-based fibers involved encapsulating the liquid metal within an elastomer. However, this encapsulation method proves to be unsuitable for various electronics applications due to the insulation of the conducting electrode by the elastomer. In contrast, our approach involves implementing the conducting electrode outside of the fiber, facilitating seamless integration with other electronic components.

Additionally, previous liquid metal fibers have faced challenges related to leakage, mainly due to mechanical instability of LM. This issue significantly hampers their integration with other electronic components. Conversely, our mechanically durable liquid metal composite ensures stable operation without any leakage.

We have revised the introduction section in the main text, incorporating a discussion on relevant previous research as references:

Revised Main Text

Moreover, conventional nano materials coating techniques, such as dip coating or spray coating, are not suitable for micro-scale LMPs.^{22,27} **Traditionally, the integration of LM or**

LMP into fiber involves encapsulating them with elastomers.^{28,29} However, this approach limits the ease of integration with electronic components and restricts their use as direct electrodes due to the presence of the encapsulation layer. Additionally, the inherent instability of LM or LMP poses a significant challenge, primarily because of the potential for leakage.

Added References

- 28 Zheng, L. J. *et al.* Conductance-stable liquid metal sheath-core microfibers for stretchy smart fabrics and self-powered sensing. *Science Advances* **7**, doi:10.1126/sciadv.abg4041 (2021).
- 29 Qu, Y. P. *et al.* Superelastic Multimaterial Electronic and Photonic Fibers and Devices via Thermal Drawing. *Advanced Materials* **30**, doi:10.1002/adma.201707251 (2018).

2. There may be various parameters, such as the fiber shape, fiber size, material properties, coating speed that could affect the output of the coating process. It is recommended that the authors conduct a parameter analysis

Answer. We appreciate the Reviewer for help comment. In response to their comment, we conducted a parameter analysis while keeping the materials and coating speed fixed, as they had already been optimized. As the Reviewer mentioned, the type and size of the fiber are important factors in determining its conductivity. Our findings indicate that the rectangular form of the fiber exhibits the highest conductivity compared to other types of fibers. Furthermore, increasing the fiber width leads to a decrease in resistance per unit length.

We have added this data in the Supplementary Information and revised the Method Section to ensure the importance of optimized coating condition.

Added Figure and caption in the Supplementary Information

Supplementary Fig. 14| Resistance of BiLMP-coated fibers. a, Resistance of BiLMP coated fiber according to type. **b,** Resistance of BiLMP coated rectangular shape fiber according to width of diameter.

To achieve reliable electrical conductivity, it is recommended to use fibers with a width of over 50 μm .

Revised Main Text

When CaLMP is coated on top of the PaLMP layer, the resulting bi-layered structure results high conductivity ($2.24 \times 10^6 \text{ S/m}$, **Supplementary Fig. 14** show the conductance of different type of fiber).

Revised Method section

A customized blade coating machine was employed to coat PaLMP films onto a fiber substrate. In order to conduct desired solution shearing, the blade was tilted to 5° and the gap between the blade and substrate was maintained at 200 μm . Prior to the solution shearing process, the fibers were attached to the glass substrate using polyimide tape and subjected to treatment with oxygen plasma (Femto Science) at 100W for 1 minute, in order to clean and enhance wettability of the surface. To ensure reliable coating and achieve electrical conductivity, we optimized the coating conditions. The process involved heating the glass substrates to 70°C . We introduced 200 μl of PaLMP ink between the blade and the substrate, which was then coated on the substrate at a rate of 0.7 mm/s. This controlled temperate and speed allowed for the efficient evaporation of the solvent during the shearing process. Following the shearing process, the PaLMP-coated fiber underwent further heating on a hot plate set at 70°C to ensure complete evaporation of any residual solvent.

3. How will the operation environment affect the performance (e.g. temperature, humidity)?

Answer. We thank the Reviewer for the helpful comment. We acknowledge that during the solution process, the facilitated evaporation of the solvent plays a critical role in achieving the desired coated LMP. As a result, factors that influence the evaporation of the solvent are of importance. Given the inherent difficulty in controlling the humidity of the coating environment, we have instead focused on controlling the temperature during the coating process. By heating the substrate to 70°C , we ensure an effective coating of LMP with reliable electrical conductivity. If the substrate is not heated, LMP film is not robustly adhered to the substrate.

Additionally, we have conducted an investigation into the influence of operating temperature on the performance of the coated fiber. Resistance measurements of the fabricated BiLMP film were carried out across a temperature range of 20°C to 50°C . It was observed that the resistance of the BiLMP film increased as the temperature rose. However, even at the highest tested temperature of 50°C , the change in resistance remained negligible. Therefore, we can conclude

that there are no significant concerns regarding the use of this film in bioelectronics applications, as it maintains stable performance across the investigated temperature range.

In response to this insight, we have added additional data and revised our Main Text and Method.

Revised Main Text

If the solvent has no acidity, such tight adhesion is not achieved and particles are easily delaminated as shown in **Supplementary Fig. 11**. Additionally, if the substrate is not subjected to heating, the chemical annealing of particles necessary for robust adhesion is not fully realized.

Revised Method section

A customized blade coating machine was employed to coat PaLMP films onto a fiber substrate. In order to conduct desired solution shearing, the blade was tilted to 5° and the gap between the blade and substrate was maintained at 200 μm. Prior to the solution shearing process, the fibers were attached to the glass substrate using polyimide tape and subjected to treatment with oxygen plasma (Femto Science) at 100W for 1 minute, in order to clean and enhance wettability of the surface. To ensure reliable coating and achieve electrical conductivity, we optimized the coating conditions. The process involved heating the glass substrates to 70 °C. We introduced 200 μl of PaLMP ink between the blade and the substrate, which was then coated on the substrate at a rate of 0.7 mm/s. This controlled temperate and speed allowed for the efficient evaporation of the solvent during the shearing process. Following the shearing process, the PaLMP-coated fiber underwent further heating on a hot plate set at 70 °C to ensure complete evaporation of any residual solvent.

Added Figure and caption in Supplementary Information

Supplementary Fig. 22| Resistance variation of BiLMP film according to temperature.

Revised Main Text

The reliable operation capacity of BiLMP coated fiber under repeated strain is demonstrated by the cycle test (**Fig. 3c**). Additionally, the long-term stability and temperature stability of the BiLMP-coated fiber are presented in **Supplementary Figs. 21 and 22**.

4. Fig. 4a-c shows the electrophysiological recording through coated fibers when the fibers in pristine (0% strain) and stretched (20% strain). More details should be provided on how the fiber was stretched by 20% when it is implanted in the mouse brain.

Answer. We appreciate the Reviewer for the comment regarding the explanation on the stretching condition of implanted fiber. In detail, we implanted the fiber in CA1 brain region and fixed them to the mouse cranium with dental cement. To verify stable electrophysiological recording under a stretched state, the upper part of the fiber was elongated (20% strain, initial length 5mm) using a three-dimensional manipulator of stereotaxic instruments. To avoid undesired artifacts, all recording instruments have been fixed except for fiber itself. In the experiment, the signals quality that our stretchable fiber records as an electrode or connection is preserved under extreme mechanical deformations.

We have added more details in Method section in Main Text.

Revised Method section in Main Text

In vivo monitoring applications. The optical stimulation is 10mW at the tip and 5ms pulse-width. In the experiment to verify the stable electrophysiological recording during stretching deformation, we fixed the fibers to the mouse cranium with dental cement, and upper part of them were elongated (20% strain, initial length 5mm) with a three-dimensional manipulator of stereotaxic instruments. To avoid undesired artifacts, all recording instruments have been fixed except for fiber itself.

5. When the soft and stretchable fiber was implanted into the brain, did the authors encounter any problems (difficulties) with insertion? What are the size and stiffness of the fiber probes?

Answer. We appreciate the Reviewer’s question about the insertion problem. As you imagined, BiLMP-coated recording fibers (BiLMP-R-fibers) are inserted with difficulty owing to its softness and thin form factor (~ 200 μ m diameter). To address the buckling effect, we coated the fibers with bioresorbable material PEG (polyethylene glycol), which helped the insertion process. We have revised contents about PEG coating for insertion in Method section.

In addition to it, we also added the information about the size and stiffness of all used fibers in the experiment for understanding of readers. Especially, the bending stiffness information of all the fibers in the 0.1~10Hz range of frequency were provided in Supplementary information.

Revised Method section in Main Text

***In vivo* monitoring applications. To insert fiber without buckling effect, we coated BiLMP-R-fibers with polyethylene glycol (MW 6000, Sigma Aldrich).**

Revised Main Text

Due to the adaptability of meniscus-guided coating, BiLMP film can be coated on various types of fibers such as stretchable multifilament fiber (polyurethane (PU)PU fiber), commercial threads, and thermally drawn multifunctional polymer fiber, and soft threads (Stiffness of each fiber type devices is presented in **Supplementary Fig. 27**).

Added Figure in the Supplementary Information

Supplementary Fig. 27| Bending stiffness of BiLMP fibers. Bending stiffness of stainless steel, bare/coated BiLMP-M-fiber, and bare/coated BiLMP-R-fibers.

6. Please clarify what the sampling rates are for the neural recording (such as in Fig. 4d and Supplementary Fig. 24d). Also, scale bars should be added in Fig. 4f and 4i.

Answer. We appreciate the Reviewer for careful review. The neural signals for optically evoked potential and spontaneous activity were recorded with sampling rate of 24414Hz. And the ECoG signals were recorded with sampling rate of 3052Hz.

We have added information of recorded neural signal sampling rate in Method section in Main Text and have added scale bars in Fig. 4f and 4i.

Revised Method section in Main Text

***In vivo* monitoring applications.** All application procedures on the mouse model were performed with approval by the Institutional Animal Care and Use Committee (IACUC) of KAIST. The cranial window of a 2mm x 2mm scale was generated in a mouse skull with the surgical dental drill (centered on coordination AP: 2mm, ML: 2mm), and the recording was conducted with a filtering frequency range of 3~300Hz and sampling rate of 3052Hz. All neural signals were recorded using a LABRAT instrument The devices were inserted on a stereotaxic frame (Digital Stereotaxic Instrument 68025, RWD Life Science Corp., Shenzhen, China). The extacellular electrophysiological recording was conducted with a filtering frequency of 300~5000Hz and sampling rate of 24414Hz. For chronic implantation,

Modified figure and caption

Fig.4|Neural recording and stimulation with BiLMP-coated fibers for multifunctional 1D bioelectronics.

a, f, Cross sectional image of BiLMP-multifunctional (extracellular recording, optical stimulation, and drug delivery) fiber device (BiLMP-M-fibers). **Scale bar represent 100 μm .** **i**, Illustration (left) and image (right) of sciatic nerve electrical stimulation using BiLMP-based stimulation thread devices (BiLMP-S-threads) with simultaneous EMG recording. **Scale bar represent 10 mm.**

7. In the biocompatibility test (supplementary Fig. 25), the authors claimed there is no immune response based on the comparison of the histology data with/without devices. It is hard to see the location of the devices in the images. Please circle or indicate the location of the devices. The image quality is also not high.

Answer. The devices were located subdural region with conformable contact, and we added dashed yellow line to indicate the location in the revised manuscript. The quality of confocal images are also updated following the reviewer’s suggestion.

Modified Figure and caption in the Supplementary Information

Supplementary Fig. 33 In vivo biocompatibility of BiLMP electrode. **a**, Surgical procedure for in vivo biocompatibility test. **b**, Representative confocal images indicating foreign body responses of mouse brain cortex with/without (left/right) BiLMP-based devices. **Dashed yellow lines indicate the location of devices.** Scale bars represent 500 μm . **c**, Expression of GFAP and CD68 after 2 weeks and 2 months implantation. (n=8)

Reviewer #3

This manuscript reports the shearing-based deposition of polymer-attached liquid metal particles (PaLMP) and CNT-attached LMP (CaLMP) to create bilayer LMP (BiLMP) on polymer fibers. The BiLMP-coated fibers exhibited the electrical conductivity (2.24×10^6 S/m), stretchability (150%), and biocompatibility. The BiLMP-coated fibers also exhibited stable conductance under strain with the lowest gauge factor among conductive filler-coated fibers in literature. The applications of BiLMP-coated fibers such as electrical circuit, smart clothes, and stretchable bioelectrodes are demonstrated. However, the chemical mechanism, oxidation study, and detailed electrical characterization are missing in the manuscript. The novelty in the fiber synthesis itself is doubtful although fancy demonstrations are carried out. The following concerns should be addressed properly before considering this work for publication in Nature Communications.

Answer. We appreciate the Reviewer for the careful review of our work. Point-by-point responses are attached below.

Previous studies have indicated that the use of rigid conductive fillers in fabricating conductive fiber electrodes results in a high gauge factor, as predicted by percolation theory. This characteristic has hindered their application as electrical interconnects or electrodes, despite the significant demand for such applications. To address this issue, we have developed a novel coating technique utilizing liquid metal particles as deformable conductive fillers. This allows stable conductance under strain which allow the operation of multiple stretchable electronic applications with high fidelity.

Furthermore, mechanical durability is important for fiber electrodes. However, conventional liquid metals are susceptible to external stimuli due to their thin oxide and polymer encapsulation layers. As a result, they can only be utilized by injection into hollow tubes, rendering them unsuitable for use as fiber electrodes (*Science Advances*, 7(22), p.eabg4041). In our approach, we enhance mechanical durability by coating mechanically robust carbon nanotube (CNT)-attached liquid metal onto polymer-attached liquid metal particles.

To emphasize the primary novelty of our work, we have made revisions to the introduction sections in the Main Text.

Revised Main Text

Moreover, conventional nano materials coating techniques, such as dip coating or spray coating, are not suitable for micro-scale LMPs.^{22,27} Traditionally, the integration of LM or LMP into fiber involves encapsulating them with elastomers.^{28,29} However, this approach limits the ease of integration with electronic components and restricts their use as direct electrodes due to the presence of the encapsulation layer. Additionally, the inherent instability of LM or LMP poses a significant challenge, primarily because of the potential for leakage.

1. There is no explanation of why CNTs are functionalized with Pt nanoparticles prior to the LMP functionalization for the synthesis of CaLMP. What happens without the Pt coating? How is the LMP attached to CNT (chemical bonding mechanism)? What is the exact chemical bonding mechanism between PSS and LMP?

Answer. We thank the Reviewer for the helpful comment. One important characteristic of liquid metal is its affinity to metal surfaces (*Nat Commun* 11, 1002 (2020), *Nano Letters* 19, 4866 (2019)). In our study, the functionalization of Pt on carbon nanotubes (CNTs) plays a crucial role in promoting the integration of liquid metal particles (LMP) and CNTs. In the absence of Pt, the LMP and CNTs do not mix well, resulting in the observation of segregated layers in the solution. CNTs used in our study were functionalized with carboxyl groups through nitric acid treatment. Recent studies have reported that carboxyl groups can form bonds with gallium-based liquid metals through redox reactions (*Compos. B. Eng.* 239, 109961 (2022), *Nat Commun.* 10, 3514 (2019) *J. Am. Chem. Soc.* 144 (15), 6779 (2022)).

We have added a photograph of the LMP-CNT solution with and without functionalization with Pt in Supplementary Information and added related References.

The interaction between polystyrene sulfonate (PSS) and LMP is predominantly governed by electrostatic forces. The liquid metal particles possess a positive charge on their surface oxide layer, while PSS is negatively charged as a polymer. This electrostatic attraction enables effective coating of the PSS onto the surface of the liquid metal particles, thereby promoting their stabilization and dispersion in the solution. The observed variance in zeta potential, as discussed in Supplementary Fig. 3, further supports this tendency.

Furthermore, it is worth noting that the electrostatic interaction between PSS and LMP is reduced in the presence of a polar organic solvent. This reduction in electrostatic interaction leads to the delamination of PSS from the LMP, allowing for their electrical activation through partial merging. This process bears similarities to the phenomenon of PSS segregation-induced PEDOT bridging observed in PEDOT:PSS systems (*ACS Appl. Mater. Interfaces* 8, 302–310 (2016)). As depicted in Supplementary Fig. 17, the initial state shows PSS molecules covering the surface of the LMP. However, upon immersion in DMSO, the PSS molecules detach from the LMP and segregate from each other.

In summary, the electrostatic interaction between PSS and LMP plays a crucial role in their initial stabilization and dispersion. The introduction of a polar organic solvent diminishes this interaction, resulting in the delamination of PSS from the LMP and promoting electrical activation through partial sintering, which is also discussed below.

We have added EDS data in Supplementary Information.

Added Figure and caption in Supplementary Information

Supplementary Fig. 16| Photograph of CNT-LMP solution with and without functionalization of Pt.

The affinity of gallium-indium LM with metal facilitates its incorporation with Pt-functionalized CNTs. In the absence of Pt functionalization, CNTs and LM do not integrate well, leading to the formation of separate layers in the solution.

Supplementary Fig. 3| Schematic illustration of fabrication process of PaLMP ink.

To fabricate the PaLMP ink, PSS and bulk liquid metal are inserted in diluted acetic acid (AA) aqueous solvent (5 vol.%). To make uniform liquid metal micro particles, an acoustic field was applied for 5 min using a tip sonicator. During this process, PSS are attached to the liquid metal particles and are stabilized in the solution. **The observed variance in Zeta potential values further supports this phenomenon. Specifically, the Zeta potential value of the LMP in diluted AA is measured at +53.2 mV, indicating a positively charged surface, while the PaLMP exhibits a Zeta potential value of -21.1 mV, suggesting a negatively charged surface. These distinct Zeta potential values serve as confirmation of the electrostatic interaction between the LMP and the negatively charged polymer.**

Added Figure and caption in Supplementary Information

Supplementary Fig. 17| Energy-dispersive X-ray spectroscopy (EDS) of PaLMP. a, EDS mapping of carbon element in coated PaLMP. **b,** EDS mapping of carbon element in PaLMP after immersion in DMSO.

Upon coating the PaLMP, a carbon peak was detected on the surface of the LMP, indicating the presence of the PSS polymer that enveloped the LMP, which served as an electrical insulator. However, upon immersion in DMSO, we observed the detachment and segregation of carbon from the PSS polymer on the LMP surface, similar to the reaction observed in PEDOT:PSS when exposed to DMSO (Lee et al., 2016). The introduction of a polar organic solvent weakens the electrostatic interaction between the PSS and LMP, leading to the detachment of PSS from the LMP surface. This delamination of the polymer enables the establishment of electrical connections between the LMP particles.

Lee, I., Kim, G. W., Yang, M., & Kim, T. S. (2016). Simultaneously enhancing the cohesion and electrical conductivity of PEDOT: PSS conductive polymer films using DMSO additives. *ACS applied materials & interfaces*, 8(1), 302-310.

2. Fig. 3a. It is strange that the electrical conductivity of CaLMP is significantly smaller than that of PaLMP. Is this the conductivity of entire fiber or coating layer only? This should be clarified in the entire manuscript. Why is the conductivity of BiLMP higher since it is only the double layer of CaLMP and PaLMP? What is the thickness of each layer? What is the concentration of each component (e.g., CNT concentration in the entire composite fiber)? What is the concentration of EGaIn? What is the measured conductivity of EGaIn? Apparently, the CNT did not increase the conductivity. What is the major mechanism of achieving high conductivity of BiLMP-coated fiber? How did they measure the conductivity precisely?

Answer. We thank the Reviewer for the helpful comments.

(It is strange that the electrical conductivity of CaLMP is significantly smaller than that of PaLMP)

During the bottom layer coating process using PaLMP, an acidic solvent induces partial merging of the LMP at the bottommost layer by reducing the polymer and oxide layer. This partial sintering enables the establishment of electrical conductivity between LMPs. To provide comprehensive evidence of the partial sintering of PaLMP, we have included this data in the Supplementary Information and we have revised the related detail information in the Method section.

(Is this the conductivity of entire fiber or coating layer only?, What is the measured conductivity of EGaIn?, How did they measure the conductivity precisely?, What is the thickness of each layer?)

In the case of conductivity, we measure the conductivity of each film on the plane substrate (TPU) to eliminate geometrical factors. The reported conductivity of EGaIn is 3.4×10^6 S/m. The coated PaLMP and CaLMP layers had approximate thicknesses of 15 μm and 5 μm , respectively. However, during the coating of the top layer, further annealing induces the adhesion between two layers; therefore, the precise thickness of each layer is hard to measure as presented in Supplementary Fig 13. We have revised the Method section to clarify our data

(What is the concentration of each component (e.g., CNT concentration in the entire composite fiber)? What is the concentration of EGaIn?)

Determining the precise concentration of each component within the coated fiber proved to be a challenge due to the application of BiLMP film onto the commercially available fiber. In particular, the concentration of CNT and polymer relative to EGaIn is estimated to be around 0.3% and 7% by weight, respectively. We have made revisions to the Method section.

(What is the major mechanism of achieving high conductivity of BiLMP-coated fiber?)

The presence of an oxide and polymer insulation layer on the PaLMP film requires an additional electrical activation process to establish a complete electrical connection and achieve high electrical conductivity in the BiLMP structure. This is because the initial state of PaLMP is not fully electrically connected. The key mechanism for achieving high conductivity involves further annealing of the PaLMP layer during the coating of the CaLMP layer.

Here, we employed DMSO, a widely recognized polar organic solvent, as a solvent of CaLMP. During the coating process of the CaLMP layer, DMSO plays a crucial role in inducing the delamination of PSS from LMP. This is achieved by reducing the interaction between the polymer and LMP, facilitating the separation of PSS from the LMP. This process bears similarities to the phenomenon of PSS segregation-induced PEDOT bridging observed in PEDOT:PSS systems (*ACS Appl. Mater. Interfaces* 8, 302–310 (2016)).

The delamination of PSS enables the sintering of the LMPs, which establishes electrical connections between them. This delamination and sintering process serves as the primary mechanism responsible for the high conductivity achieved in the BiLMP structure, rather than the inclusion of a CNT network. However, it is important to note that CNTs contribute to enhancing the mechanical durability of the structure.

To substantiate the claim regarding the detachment of PSS after immersion in DMSO, we have included EDS data in the Supplementary Information.

Added Figure and caption in Supplementary Information

Supplementary Fig. 10| Chemical annealing of PaLMP. **a**, Schematic illustration of chemical annealing by acid during solution shearing. **b**, Bottom optical microscope image of coated PaLMP with acid. **c**, Gallium X-ray Photoelectron Spectroscopy (XPS) spectrum of PaLMP with acid. **d**, Bottom optical microscope image of coated PaLMP without acid. **e**, Gallium XPS spectrum of PaLMP without acid.

The process of suspension shearing involves evaporation of the solvent at an enlarged surface area in the meniscus, facilitated by heating. The ink used in this process is a diluted acetic acid (AA) mixed with water. As the boiling point of AA is higher than that

of water, the acidity of the solvent increases during suspension shearing, which is evident from Supplementary Fig. 9. This increased acidity induces the chemical reduction of the surface oxide layer of the PaLMP. As the oxide is partially removed, the particles sinter, and the oxide layer naturally reforms at the surface. During this process, an oxide layer is formed at the interface between the LMP and the substrate, which acts as an adhesive layer, resulting in robust adhesion.

The bottom layer of the PaLMP film, as shown in the bottom OM image, exhibits a sintered particle morphology, indicating the successful particle sintering process. Additionally, the presence of a Gallium element peak in the XPS further supports our findings of consumption. On the other hand, in cases where there is no particle sintering, only an oxide peak is observed in the XPS spectra

Added Figure and caption in Supplementary Information

Supplementary Fig. 17| Energy-dispersive X-ray spectroscopy (EDS) of PaLMP. a, EDS mapping of carbon element in coated PaLMP. **b,** EDS mapping of carbon element in PaLMP after immersion in DMSO.

Upon coating the PaLMP, a carbon peak was detected on the surface of the LMP, indicating the presence of the PSS polymer that enveloped the LMP, which served as an electrical insulator. However, upon immersion in DMSO, we observed the detachment and segregation of carbon from the PSS polymer on the LMP surface, similar to the reaction observed in PEDOT:PSS when exposed to DMSO (Lee et al., 2016). The introduction of a polar organic solvent weakens the electrostatic interaction between the PSS and LMP, leading to the detachment of PSS from the LMP surface. This delamination of the polymer

enables the establishment of electrical connections between the LMP particles.

Lee, I., Kim, G. W., Yang, M., & Kim, T. S. (2016). Simultaneously enhancing the cohesion and electrical conductivity of PEDOT: PSS conductive polymer films using DMSO additives. *ACS applied materials & interfaces*, 8(1), 302-310.

Revised Main Text

The reason for such high conductivity can be attributed to the fact that CaLMP is in a dimethyl sulfoxide (DMSO)-based solution (Detail information regarding ink for solution process is presented in **Supplementary Figs. 15 and 16.**), and when PaLMP is exposed to DMSO, the detachment of PSS polymer from LMP and consequent cohesion of particles is likely to be induced (**Supplementary Figs. 17 and 18**). This effect is similar to that of the phase separation of PEDOT:PSS under DMSO exposure, which has been previously studied.^{49,50}

Revised Method section

Preparation of PaLMP ink. ... at a 30% amplitude for a duration of 30 minutes. **Following the tip sonication step, additional acetic acid (10 vol.%) was added to facilitate chemical sintering.**

Characterization.

... a stand with a motor (Mark-10), and a customized manual strain machine were utilized. **To eliminate any potential geometrical factors, the conductivity of each layer is measured by coating the films on plane TPU substrates.**

3. Figure 1a schematic is confusing. How many BiLMP-coated fibers in the bundle? Each fiber has bi-layer coating? What fills the gap between fibers?

Answer. We appreciate the Reviewer's feedback. In Figure 1a, we initially referenced a commercialized fiber that consisted of several bundles. However, we understand that this depiction might have caused confusion among readers, as it does not represent the typical form of a fiber. To address this concern and improve clarity, we have made modifications to Figure 1a by illustrating a cylindrical fiber, which is the representative shape of a fiber.

We coated our BiLMP on square-shaped fiber, cylindrical fiber, and bundle fiber. In the case of bundle fiber, gap is filled with LMP as presented in Figure 2e.

We have modified the schematic illustration of Figure 1a.

Revised Figure in Main Text

4. The authors claimed that the oxide layer of the LMPs was removed by acetic acid and reformed after coating, leading to better adhesion between particles and substrate. How did the authors confirm the removal and reformation mechanism of the oxide layer other than the schematic in Supplementary Fig.10. Any spectroscopic evidence?

Answer. We appreciate the helpful comment from the Reviewer. Previous research has emphasized the significance of in situ oxide layer formation in facilitating the adhesion of liquid metal to substrates (doi.org/10.1002/admt.202000070, doi.org/10.1002/adem.201900400). However, in the case of our PaLMP, microparticles already possess a polymer-covered oxide layer on their surfaces. Consequently, when these particles are applied to a substrate, they tend to stack on top of each other rather than undergoing chemical sintering or forming a robust adhesion. As a result, the adhesion between the particles is weak and easily delaminates, as illustrated in Supplementary Figure 11.

To achieve strong adhesion, we have developed a novel process that involves extruding the bulk liquid metal by removing the outer oxide layer through acid treatment. The extruded liquid metal naturally forms a new oxide layer, which acts as an adhesion layer between the substrate and the particles. The robust adhesion between the particles can be observed in the bottom optical microscopy image. As depicted in Supplementary Figure 10, PaLMPs dispersed in an acidic ink undergo sintering and firmly adhere to the glass substrate. In contrast, PaLMPs without acid treatment only exhibit a stacked morphology.

Furthermore, we have obtained XPS data of the bottom layer with and without acid treatment. Notably, only in the case of acid treatment, a small amount of gallium peak is observed alongside the gallium oxide peak. This finding confirms the extrusion of liquid metal and the subsequent reformation of the oxide layer.

To provide comprehensive evidence of the sintering and adhesion process, we have included this data in the Supplementary Information, which offers visual and chemical confirmation of the phenomenon.

Added Figure and caption in Supplementary Information

Supplementary Fig. 10| Chemical annealing of PaLMP. **a**, Schematic illustration of chemical annealing by acid during solution shearing. **b**, Bottom optical microscope image of coated PaLMP with acid. **c**, Gallium X-ray Photoelectron Spectroscopy (XPS) spectrum of PaLMP with acid. **d**, Bottom optical microscope image of coated PaLMP without acid. **e**, Gallium XPS spectrum of PaLMP without acid.

The process of suspension shearing involves evaporation of the solvent at an enlarged surface area in the meniscus, facilitated by heating. The ink used in this process is a diluted acetic acid (AA) mixed with water. As the boiling point of AA is higher than that of water, the acidity of the solvent increases during suspension shearing, which is evident from Supplementary Fig. 9. This increased acidity induces the chemical reduction of the surface oxide layer of the PaLMP. As the oxide is partially removed, the particles sinter, and the oxide layer naturally reforms at the surface. During this process, an oxide layer is formed at the interface between the LMP and the substrate, which acts as an adhesive layer, resulting in robust adhesion.

The bottom layer of the PaLMP film, as shown in the bottom OM image, exhibits a sintered particle morphology, indicating the successful particle sintering process. Additionally, the presence of a Gallium element peak in the XPS further supports our findings of consumption. On the other hand, in cases where there is no particle sintering, only an oxide peak is observed in the XPS spectra.

Revised Main Text

Reformation of the oxide layer with the extruded LM near the surface allows the robust adhesion between particles and substrate as illustrated in region ③ (see **Supplementary Fig. 10** for robust adhesion of the bottommost layer).^{45,46}

Added Reference

46 Neumann, T. V. & Dickey, M. D. Liquid Metal Direct Write and 3D Printing: A Review. *Advanced Materials Technologies* **5**, doi:10.1002/admt.202000070 (2020).

5. Fig. 3d. The coating is only the physical adsorption. The scotch tape tests only the adhesion of the skin layer. The adhesion of PaLMP was bad. How stable is the adhesion of the bilayer on the polymer fiber?

Answer. We appreciate the Reviewer for the helpful comment. As mentioned earlier, the chemical annealing of PaLMP leads to robust adhesion. However, the very top layer covered with polymer remains weak, resulting in its delamination and the subsequent rupturing of liquid metal particles after peel-off with scotch tape, as shown in Supplementary Fig. 22.

To address this issue, we introduced a layer of CNT-coated LMP on top of the film during the formation of BiLMP. The presence of CNT enhances the mechanical stability of the structure, preventing delamination or rupturing of the liquid metal layer.

To provide visual evidence of the impact of CNT on the structure, we have included SEM images of the film after peel-off with scotch tape in the Supplementary Information. The corresponding modifications have also been made in the Main Text to reflect this additional information.

6. The delamination of bare liquid metal particles under strain is provided in the supplementary Fig. 11. Do the BiLMs delaminate under strain? This should be compared under the same condition.

Answer. We thank the Reviewer for the helpful comment. The robust adhesion between the bottom layer and substrate is a critical factor in achieving successful integration of the BiLMP film onto the stretchable substrate. This strong adhesion is accomplished through annealing with an acidic solvent. The annealing process ensures a firm bond between the bottom layer and the substrate, which helps prevent delamination even under strain. This is in contrast to PaLMP films (bottom layer) prepared without the use of an acidic solvent, which may be more prone to delamination under similar conditions.

We have included the BiLMP film under strain compared to PaLMP coated film without acidic solvent in the Supplementary Information.

Added Figure and caption in Supplementary Information

Supplementary Fig. 11| Adhesion between LMP and substrate a, Photograph of delamination of PaLMP film generated with solution without acetic acid. **b**, Photograph of BiLMP film under strain.

As discussed in Supplementary Fig. 10, the annealing process using an acidic solvent during the coating stage is essential for achieving robust adhesion of the PaLMP film to the substrate. This annealing process promotes the reformation of an adhesive oxide layer between the particles and the substrate, enhancing the adhesion strength. In contrast, when the chemical annealing process is omitted, the PaLMP film retains a polymer layer on its surface. This incomplete annealing results in insufficient adhesion, leading to easy delamination of the particles.

When the bottom PaLMP layer is firmly adhered to the substrate through annealing with an acidic solvent, the integration of the BiLMP film with a stretchable substrate becomes robust. Consequently, even under strain, the BiLMP film exhibits resistance to delamination. The acidic solvent facilitates the formation of a strong and stable adhesive layer, contributing to the enhanced adhesion properties of BiLMP.

7. In Fig. 3b, the authors explained the increase in resistance of the CaLMP-coated fiber is attributed to the disruption of rigid CNT percolation with strain. Then, why is there no such increase in resistance in BiLMP since its top layer is CaLMP? Provide the mathematical resistance modeling of this process.

Answer. We appreciate the Reviewer for the helpful comment regarding resistance modeling. We can model the BiLMP structure as a parallel connection of resistors. According to the principles of parallel resistors, the overall resistance is determined by the resistor with the smallest resistance value. In our case, the sintered PaLMP layer exhibits a lower resistance and remains stable under strain. This characteristic ensures that the overall resistance of the BiLMP structure does not experience a significant increase under strain.

To provide a clearer understanding of this modeling concept, we have included a mathematical modeling section and a corresponding schematic illustration in the Supplementary Information.

Revised Main Text

As a result, our PaLMP and BiLMP film shows almost constant resistance under the application of strain as shown in **Fig. 3b** (Further discussion regarding resistance change of BiLMP film is presented in **Supplementary Fig. 20**).

8. In the electrical property explanation (line 177-185), the authors claimed that they overcome the oxide layer formation of LMPs by the synthesis of BiLMP. On the contrary, in line-142, they explained that the reformation of the oxide layer in the PaLMP enhances the adhesion to the substrate. The two claims are contradicting. The oxide formation and removal also need to be experimentally confirmed.

Answer. We appreciate the Reviewer for their careful review of our work.

[Conventional challenge of oxide formation on LMP]

One of the main challenges associated with liquid metal particles for implementing on electronics is the naturally formed insulating oxide layer on their surfaces. Numerous efforts have been made to develop effective methods for electrical activation of these particles by removing the oxide layer. Conventionally, mechanical rupturing of particles through strain or stress has been widely used; however, this approach reintroduces the issue of bulk liquid metal such as leakage, instability, and fluidity due to the re-merging of particles. In our study, we have successfully developed a non-mechanical method for electrical activation using a polar organic solvent, resulting in the creation of a mechanically durable and electrically conductive bi-layer composite.

[Reformation of oxide layer as adhesive layer]

While the oxide layer acts as an electrical insulator, it can also function as an adhesive layer. Originally, PaLMP covered with polymer and oxide, its adhesion is not enough to sustain mechanical deformation; therefore, severe delamination is observed under application of strain. Therefore, the reformation of the oxide layer between the particles and the substrate is crucial to achieve robust adhesion. The removal of the oxide layer at the bottommost layer and the sintering between particles can be visually confirmed through the bottom optical microscopy (OM) image, especially when acid is present, as shown in Supplementary Figure 10.

Additionally, X-ray photoelectron spectroscopy (XPS) data further support our hypothesis regarding the removal of the oxide layer and the sintering process.

To provide a comprehensive understanding of these phenomena, we have included the bottom OM image and XPS data in the Supplementary Information. Moreover, corresponding revisions have been made in the Main Text and reference paper were added to clarify our claim.

Added Figure and caption in Supplementary Information

Supplementary Fig. 10| Chemical annealing of PaLMP. **a**, Schematic illustration of chemical annealing by acid during solution shearing. **b**, Bottom optical microscope image of coated PaLMP with acid. **c**, Gallium X-ray Photoelectron Spectroscopy (XPS) spectrum of PaLMP with acid. **d**, Bottom optical microscope image of coated PaLMP without acid. **e**, Gallium XPS spectrum of PaLMP without acid.

The process of suspension shearing involves evaporation of the solvent at an enlarged surface area in the meniscus, facilitated by heating. The ink used in this process is a diluted acetic acid (AA) mixed with water. As the boiling point of AA is higher than that of water, the acidity of the solvent increases during suspension shearing, which is evident from Supplementary Fig. 9. This increased acidity induces the chemical reduction of the surface oxide layer of the PaLMP. As the oxide is partially removed, the particles sinter, and the oxide layer naturally reforms at the surface. During this process, an oxide layer is formed at the interface between the LMP and the substrate, which acts as an adhesive layer, resulting in robust adhesion.

The bottom layer of the PaLMP film, as shown in the bottom OM image, exhibits a sintered particle morphology, indicating the successful particle sintering process. Additionally, the presence of a Gallium element peak in the XPS further supports our findings of consumption. On the other hand, in cases where there is no particle sintering, only an oxide peak is observed in the XPS spectra.

Revised Main Text

Reformation of the oxide layer with the extruded LM near the surface allows the robust adhesion between particles and substrate as illustrated in region ③ (see **Supplementary Fig. 10** for robust adhesion of the bottommost layer).^{45,46} In contrast, when the chemical annealing process is not performed, the PaLMP film remains covered with a polymer layer. As a result, the desired tight adhesion is not achieved, leading to the easy delamination of particles, as demonstrated in **Supplementary Fig. 11**

Added Reference

- 46 Neumann, T. V. & Dickey, M. D. Liquid Metal Direct Write and 3D Printing: A Review. *Advanced Materials Technologies* **5**, doi:10.1002/admt.202000070 (2020).

REVIEWERS' COMMENTS

Reviewer #1 (Remarks to the Author):

The authors addressed well the comments from the reviewers. I think the manuscript can be published as it is.

Reviewer #2 (Remarks to the Author):

The authors have fully addressed my previous concerns, and the manuscript has been significantly improved. I recommend the acceptance of the manuscript for publication.

Reviewer #3 (Remarks to the Author):

The authors responded to all the comments from this reviewer with additional experiments and characterizations. This manuscript can be accepted in Nature Communications in its current form.

Response to Reviewers' Comments

The Reviewer's comments are in **bold** and revised texts are **highlighted**.

Reviewer #1

The authors addressed well the comments from the reviewers. I think the manuscript can be published as it is.

Reviewer #2

The authors have fully addressed my previous concerns, and the manuscript has been significantly improved. I recommend the acceptance of the manuscript for publication.

Reviewer #3

The authors responded to all the comments from this reviewer with additional experiments and characterizations. This manuscript can be accepted in Nature Communications in its current form.

Answer. We appreciate the Reviewers for the careful reviews of our work.